Comparison of selective laser melting and stereolithography etching templates for guided endodontics

Zhang Ting 1
Chen Du 2
Zhang Fei 1
Xie Sijing 1
Wu Guofeng 1
Hu Qingang 1
Yan Fuhua 1 yanfh@nju.edu.cn
Tang Xuna 1 xunatang@126.com
1 Nanjing Stomatological Hospital, Affiliated Hospital of Medical School, Research Institute of Stomatology, Nanjing University , Nanjing , China
2 The Affiliated Stomatological Hospital of Nanjing Medical University, Jiangsu Province Key Laboratory of Oral Diseases, Jiangsu Province Engineering Research Center of Stomatological Translational Medicine , Nanjing , China
Abu Hasna Amjad
Electronic publication date: 2024 Jul 23
Publication date: 2024
Volume: 12
Electronic Location ID: e17646
Received 2024 Apr 1; Accepted 2024 Jun 6
Copyright: © 2024 Zhang et al.
Copyright year: 2024
Copyright holder: Zhang et al.
License: This is an open access article distributed under the terms of the Creative Commons Attribution License, which permits unrestricted use, distribution, reproduction and adaptation in any medium and for any purpose provided that it is properly attributed. For attribution, the original author(s), title, publication source (PeerJ) and either DOI or URL of the article must be cited.
License URL: https://creativecommons.org/licenses/by/4.0/

Keywords: Guided endodontics, Accuracy, Root canal treatment, Pulp canal obliteration

Funding: State Key Lab. for Novel Software Technology (Nanjing University) KFKT2018B16 Nanjing Clinical Research Center for Oral Diseases 2019060009 Jiangsu Provincial Medical Youth Talent QNRC2016119 Nanjing Medical Science and Technique Development Foundation YKK21185 This work was supported by the State Key Lab. for Novel Software Technology (Nanjing University) foundation (KFKT2018B16), the Nanjing Clinical Research Center for Oral Diseases (No. 2019060009), the Jiangsu Provincial Medical Youth Talent (QNRC2016119), Nanjing Medical Science and Technique Development Foundation (YKK21185). The funders had no role in study design, data collection and analysis, decision to publish, or preparation of the manuscript.

==============================
Background

With the increasing application of guided endodontics to treat complex root canal treatment, the entire process of root canal treatment has become more precise, reducing damage to tooth structure and improving success rates. However, due to the limitations of the operating space, the use of guided endodontic templates in posterior root canal treatment is less common. This study aims to compare the accuracy and reliability of selective laser melting (SLM) and traditional stereolithography etching (SLA) guided endodontic templates for posterior root canals, providing better treatment strategies for posterior root canal treatment.

Methods

The teeth were randomly assigned to either SLM or SLA group. Preoperative cone-beam computed tomography (CBCT) and a three-dimensional (3D) scanner were used to establish the 3D root canal system and the accurate occlusal models of the teeth. The virtual access to the canal access was designed using Mimics 19.0 and 3-Matic 11.0. The endodontic access was performed based on either SLM or SLA templates. The accuracy of endodontic preparation was measured in three-dimensions by calculating deviations from planned accesses. The template height and tooth substance loss rates in each group were measured.

Results

SLM-guided templates have a low average deviation at the entry point and apical portion of the bur of total posterior teeth (including premolars and molars) and individual molars (P < 0.05). Moreover, there was a significant difference in angular deviations and height of template in total posterior teeth and individual molars (P < 0.05). The mean substance loss rate of the SLA group was slightly greater than that of the SLM group, but the difference was not statistically (P > 0.05).

Conclusions

SLM-guided endodontics provides a more predictable and precise location of root canal orifice for the treatment of posterior teeth.

graphical abstract

Introduction

Establishing root canal pathways is the first step in root canal treatment (RCT). However, in many cases, pulp canal obliteration (PCO) occurs in the root canal system, blocking the original pathways, complicating the treatment, reducing the success rate, and relying on experts and being time-consuming in clinical practice (McCabe & Dummer, 2012; Wei et al., 2023). Lifelong dentin deposition can lead to partial or complete obliteration of the pulp chamber and root canal (Chaniotis, Sousa Dias & Chanioti, 2024). Research has shown that PCO mainly occurs in molars, especially the first molars (Sener, Cobankara & Akgunlu, 2009). Due to limitations in visibility and operating space, root canal treatment of posterior teeth is challenging. However, obliteration in the posterior pulp chamber is relatively prominent. Excessive damage to healthy tooth tissue, including lateral penetration, can occur when exploration of these obliterated canals during RCT (McCabe & Dummer, 2012; Oginni, Adekoya-Sofowora & Kolawole, 2009).

Many investigations have suggested that three-dimensional (3D)-printed access guides are an efficient and safe method of addressing challenging endodontic cases, improving chemo-mechanical debridement, and protecting the tooth structure (Anderson, Wealleans & Ray, 2018; Llaquet Pujol et al., 2021). Additionally, clinical studies have confirmed the safety and reliability of guided endodontics as an auxiliary positioning method for anterior teeth (Buchgreitz, Buchgreitz & Bjorndal, 2019a; Connert et al., 2019; Fonseca Tavares et al., 2018; Zehnder et al., 2016). Recently, some trials have used guided endodontics for the treatment of posterior teeth (Lara-Mendes et al., 2018; Maia et al., 2019). However, the literature is limited, particularly regarding guided endodontics access in molars because of the limited space for the template and drill (Santiago et al., 2022). Moreover, the overall height of the traditional guided endodontic templates restricts their use in the molar area. In most cases, traditional guided endodontic templates are made using materials such as polylactic acid, which are thick, unstable, and inaccurate. Furthermore, the templates require additional metal sleeves. Hence, there is a risk of separation between the sleeve and the template (Wolters et al., 2017). The laser-melting templates to treat calcified root canals in anterior teeth obtained good results by the same research group as this study (Zhang et al., 2020). To ensure adequate space during posterior RCT, this study employed laser-melting templates to treat the posterior root canals, further reducing the height of the template and difficulty in performing RCT.

A laser-melted digital metal guide template was designed to overcome the disadvantages of the existing guide templates. Subsequently, in order to evaluate the accuracy of selective laser melting (SLM) guided endodontics template, a comparison was made in vitro between the new SLM guided endodontic template and the traditional stereolithography (SLA) template for the reliability of posterior teeth. This research aims to explore a more suitable endodontic access guide template for posterior teeth. We anticipate that through the application of SLM templates, a more reliable and denpendable method can be established for the PCO in posterior teeth.

Materials and Methods

Preparation of extracted teeth

This study collected a total of 80 premolars and molars that needed to be extracted due to periodontitis or orthodontic reasons. Teeth with the following conditions will be excluded: (1) cracks, (2) defects, (3) previously treated with endodontic treatment, (4) caries, (5) immature root apices, (6) root fractures or root resorption. Ultimately, 16 premolars and 16 molars (eight from mandible and eight from maxilla, respectively) were selected for this study. The teeth were randomly divided into two groups and encased in wax molds. Through CBCT scanning, it was found that nine molars had complete obliteration of the entire pulp cavity. The teeth were numbered and using a computer-generated random number table generator, the calcified teeth were randomly assigned to the SLM and SLA groups. The remaining teeth were also randomly assigned to the SLM and SLA groups based on a computer program-generated random number table. This study was approved by the Ethical Research Committee of Nanjing Stomatological Hospital Medical School, Nanjing University (NO. 2019NL-001) and written informed consent was obtained from all participants.

Image acquisition

Two molars and two premolars from the same quadrant were embedded in paraffin in sequence within mouth, forming a “unit”. The height of the paraffin reached the enamel-dentin junction. Eight units were analyzed using CBCT (NEWTOM VG, NEWTOM, Verona, Italy) with a voxel size of 0.125 mm. The images were saved in digital imaging and communication format in medical records. The surface morphology of each unit was obtained using a digital scanner (Trios 3, 3Shape, Copenhagen, Denmark). The data were saved in the surface tessellation language format.

Design and manufacturing process of plates

Reconstruction of root canals from CBCT images was performed using Mimics 19.0 software (Materialise, Leuven, Belgium) to obtain a three-dimensional model. The three-dimensional model was then fitted to the digital scanner’s scan images using the “multipoint registration” and “repositioning” functions. Mimics 19.0 was used to repeatedly reposition and confirm the root canal orifices on multiple CBCT slices. A three-dimensional model (model R) was extracted from the root canal orifice to the middle of the root canal to determine the ideal drilling direction. Using 3-Matic 11.0 software (Materialise, Leuven, Belgium), a 15 mm high cylinder was generated for each root canal, with a base radius of 0.78 mm (the bottom of the cylinder aligned with the middle of the root), positioned and oriented closest to model R. One such cylinder was randomly selected for each premolar, and two cylinders were randomly selected for each molar, ensuring that the chosen cylinders did not overlap with the pulp canal pathways. Based on different plate designs and manufacturing processes, the eight units were randomly allocated into SLA and SLM groups, each consisting of two maxillary and two mandibular teeth.

Traditional plate

In the 3Shape Design system (Dental System 2019, 3Shape, Denmark), the removable partial denture module was used to design the templates. The undercuts of the teeth were filled automatically to ensure smooth insertion of the templates. A “window” is designed on the occlusal surface to ensure that the entire drilling path is unobstructed. The average thickness of the template is 3 mm. The sleeve retainers were drawn above the template, leaving space for the sleeve (hollow ring-shaped cylinder used for bur guidance). The sleeve retainer and template are 3D printed as a single unit using an SLA 3D printer (NextDent 5100; 3D Systems, Rock Hill, South Carolina, USA). The finished product was a metal sleeve with an inner radius of 0.75 mm, outer radius of 1.3 mm, and height of 5 mm.

Modified plate

The standard insertion path determination, undercut filling, contour line design, and “window” operations were consistent with that of traditional guide templates. The average thickness of the template was 1 mm. Using Magics 21.0 (Materialise, Leuven, Belgium), sleeves with an inner radius of 0.78 mm, outer radius of 1.3 mm, and height of 5 mm is placed 0.2 mm above the occlusal surface, and connected to the template through pillars. The modified plates were printed using SLM (Space Traveler Tr150, Profeta, China) with a Co-Cr alloy, followed by polishing.

Root canal pathway preparation

After the template is printed, install it on the teeth. Make sure the template is not loose on the teeth. One well-trained dentist performed all the operations using a unique flat-head bur with a radius of 0.6 mm (H254LE.314.012, Komet, Germany) on the operating table, which was replaced after every four surgeries. No loupes or magnification equipment was used during the procedure, and the operator did not see the radiographs of the tooth before the operation. The operation was paused when the drill reached the designated position. Postoperatively, each unit underwent CBCT examination. By fitting the pre- and post-operative CBCT data and recording the position differences between the actual and designed paths relative to the insertion angle, the average of three measurements were calculated (Fig. 1). The prepared cavity was reconstructed in three dimensions using Mimics software, and its outline was highlighted. In 3-Matic, the direction of the three-dimensional cavity was determined by fitting a cylinder, with the actual drilling direction represented by an emphasized straight line. The lowest point of the highlighted contour line corresponded to the drilling tip, whereas the crossing point between the highlighted contour line and the straight line indicated the starting position for drilling. The Euclidean distance between the expected starting point and the actual starting point was measured using Mimics software to quantify the drilling position deviation. The same approach was used to measure the positional deviation of the drill tip. In 3-Matic, the volume of the actual drilled cavity was calculated as a 3D model and compared with the expected cylindrical volume of the drilling channel, referred to as the volume deviation. The angle difference was quantified by measuring the acute angle between the highlighted straight line and the intended drilling direction. The distance from the bottom of the guide along the expected drilling direction to the nearest tooth surface was measured, termed as drilling height. The average of the three measurements were calculated.

Figure 1 A comparison of the SLM template and SLA template of guided endodontics.

(A and B) The SLM (left column) and SLA (right column) template positioned on study model. The guide must fit snugly on the teeth with no looseness. Pictures were taken after the templates were installed (image credit: Ting Zhang). (C and D) Computer simulation of the design of two different templates for guided endodontics using the 3Shape Design system (Dental System 2019; 3Shape, Copenhagen, Denmark) (image credit: Du Chen). (E and F) Examples of posterior teeth after access cavity preparation. All procedures were carried out by a skilled dentist using a flat-headed bur with specific dimensions and changing the bur after every four surgeries. The drilling was paused when the designated position was reached. Photos were taken after preparation (image credit: Ting Zhang). (G and H) Three-dimensional reconstruction of access cavities. Postoperative CBCT scans were taken and 3-Matic 11.0 software was used to reconstruct the teeth in three dimensions (image credit: Du Chen); Locate the root canal system using SLM and SLA templates before performing CBCT examination. The sagittal (I and J) and vertical (K and L) plane of a postoperative CBCT scan showing the posterior molars (image credit: Ting Zhang).

Statistical analyses

The analysis of the deviations between the planned and prepared access cavity was performed using the independent t-test (or Mann–Whitney U test for non-parametric data). The independent t-test was used to measure the height and overall substance loss in the two groups. The normality of the distribution was analyzed using the Shapiro–Wilk test. All statistical analyses were performed using Statistical Product and Service Solutions, version 25 (SPSS, Chicago, IL, USA); P-values < 0.05 were considered statistically significant.

Results

Deviation of guided endodontics

The mean deviations in the 3D vector and the angle between the bur tip and prepared access cavity were compared during planning and after the operation. In terms of the entry point of the dental drill into the pulp chamber, the mean entry point deviation of the SLM template was significantly smaller than that of the SLA template when applied to all posterior teeth. In addition, the average entry point deviation of SLM group is significantly smaller than that of the SLA group, but there is no significant difference in the deviation of the average entry point of the premolars between the two groups. Additionally, the mean entry point deviation of the SLM templates was significantly smaller than that of the SLA templates for molars, but there is no significant difference for premolars (Table 1, Fig. 2). Measurement of the mean apical deviations of the drill tip after using both types of templates revealed that the mean apical deviation of the SLM template plate was significantly smaller than that of the SLA template for all posterior teeth and molars, whereas no significant difference was observed in the mean apical deviation between the two templates for premolars. The mean angular deviation of the SLA guide template and SLM guide template for posterior teeth overall was 2.20° (median: 1.78, interquartile range (IQR): 0.76–2.99) and 3.88° (median: 3.18, IQR: 2.52–4.95), respectively, with a significant difference between them (Table 1, Fig. 2). The average deviation angles for premolars and molars in the SLM group were 2.63° and 1.98°, respectively, while in the SLA group they were 3.59° and 4.03°, respectively (Table 1, Fig. 2). Similar to the apical deviations, a statistically significant difference was observed for posterior teeth overall and molars but not for premolars.

Table 1 The deviations between planned and prepared access cavities for SLM vs. SLA.

	SLM template	SLA template		
	Mean ± SD	Median (IQR)	Mean ± SD	Median (IQR)	P	
Entry point deviations (mm)						
Total	0.22 ± 0.1	0.20 (0.135–0.27)	0.43 ± 0.28	0.34 (0.27–0.52)	0.0001	
Premolar	0.23 ± 0.13	0.17 (0.15–0.33)	0.38 ± 0.27	0.3 (0.23–0.39)	0.0884	
Molar	0.21 ± 0.09	0.20 (0.12–0.27)	0.46 ± 0.29	0.44 (0.28–0.56)	0.0005	
Apical deviations (mm)						
Total	0.28 ± 0.17	0.22 (0.16–0.46)	0.46 ± 0.24	0.44 (0.33–0.56)	0.0089	
*Premolar	0.32 ± 0.19	0.34 (0.16–0.49)	0.35 ± 0.18	0.32 (0.19–0.53)	0.8940	
Molar	0.26 ± 0.18	0.19 (0.16–0.38)	0.52 ± 0.26	0.46 (0.39–0.64)	0.0018	
Angular deviations (degrees)						
Total	2.2 ± 1.69	1.78 (0.76–2.99)	3.88 ± 2.02	3.18 (2.52–4.95)	0.0017	
Premolar	2.63 ± 2.42	1.53 (0.54–5.4)	3.59 ± 1.77	3.20 (2.30–5.19)	0.2786	
Molar	1.98 ± 1.22	1.9 (0.87–2.99)	4.03 ± 2.17	3.05 (2.62–4.74)	0.0014	
Note:

The accuracy was measured as the mean deviation between planned and prepared access cavities at the tip of the bur in terms of distance in three dimensions and angle. (SD, standard deviation; IQR, interquartile range; P. P value; *The significance of this set of data was calculated by unpaired t-test. The rest of the data was measured by the Mann–Whitney U test).

Figure 2 Distribution of 3D vector deviation between planned and prepared access cavities at the bur for SLM vs. SLA.

The prepared cavity was reconstructed in three dimensions using Mimics software, and its outline was highlighted. In 3-Matic, the direction of the three-dimensional cavity was determined by fitting a cylinder, with the actual drilling direction represented by an emphasized straight line. The lowest point of the highlighted contour line corresponded to the drilling tip, whereas the crossing point between the highlighted contour line and the straight line indicated the starting position for drilling. The Euclidean distance between the expected starting point and the actual starting point was measured using Mimics software to quantify the drilling position deviation. The same approach was used to measure the positional deviation of the drill tip. (A and B) The schematic diagram of entry point deviations for SLM and SLA guide templates. (D and E) The schematic diagram of apical deviations for SLM and SLA guide templates. The angle difference was quantified by measuring the acute angle between the highlighted straight line and the intended drilling direction. (G and H) The schematic diagram of angular deviations for SLM and SLA guide templates. 3D vector deviation between planned and prepared access cavities at the entry point (C) and the apical (F) of the bur for SLM vs. SLA group. (I) Angular deviation between planned and prepared access cavities of the bur for SLM vs. SLA group. (**P < 0.01; ***P < 0.001; ns: not significant) (Image credit: Du Chen).

The height of templates

The distance from the top of the template to the surface of each tooth was measured in this study. The height of the SLM template was significantly smaller than that of the SLA template, with a statistically significant difference (P < 0.05). The template heights of the premolars were similar (Fig. 3).

Figure 3 The height of SLM and SLA template of guided endodontics.

After designing the SLM and SLA templates, the 3matic software was used to measure the distance from the bottom of the guide to the nearest tooth surface along the intended drilling direction. The vertical distance from the bottom of the template to the nearest tooth surface along the intended drilling direction was measured. Diagram illustrating the vertical distance measurement from the top of the SLM (A) and SLA (B) templates to the tooth surface. The height of SLM and SLA template (C) (*P < 0.05; **P < 0.01; ns: not significant) (Image credit: Du Chen).

Management of PCO and substance loss rate of guided endodontics

The CBCT images of the experimental teeth revealed severe PCO in five teeth in the SLM group and four teeth in the SLA group. These PCO teeth did not exhibit obvious pulp chambers on CBCT, with varying degrees of narrowing and obliteration in the root canal spaces. All the endodontic access pathways were successfully established.

Overall, the mean substance loss rates using the SLM and SLA templates were 1.12 and 1.20, respectively. The mean substance loss rate was 1.10 in the SLM group and 1.14 in the SLA group for premolars. For molars, the mean substance loss rate were 1.15 in the SLM group and 1.25 in the SLA group (Fig. 4). There was no statistically significant difference in material loss rates between the two groups.

Figure 4 The overall substance loss of SLM and SLA template of guided endodontics.

The difference of substance loss between planned and prepared access cavities for SLM (A) vs. SLA (B). Using 3-Matic software, the volume of the actual drilled cavities was calculated in the form of a 3D model, and compared to the expected cylindrical volume of the drilling channel as volume discrepancy. The rate of substance loss of SLM and SLA template (C). (ns: not significant) (Image credit: Du Chen).

In the process of using the SLA template, there were instances of loosening and detachment between the metal sleeves and the SLA template, with three sleeves detached.

Discussion

Minimally invasive endodontics (MIE) has become a trend in endodontic therapy over the past few decades. Compared with conventional cavity access preparation, guided endodontics is a more predictable and expeditious method for locating and dredging calcified root canals, significantly reducing substance loss (Connert et al., 2019; Moreno-Rabie et al., 2020). This procedure is effective in reducing the incidence of tooth fractures after treatment (Silva et al., 2022). In this study, the tooth substance loss was compared between the two groups under guided endodontic therapy, and it was found that neither group showed excessive destruction of tooth tissues. Nevertheless, a slightly smaller substance loss was observed in the SLM group than in the SLA group. Both guided endodontic methods can be used for minimally invasive endodontic treatment of posterior teeth.

However, many factors can lead to deviations and affect the therapeutic outcomes in the treatment of posterior teeth through guided endodontics, especially with SLA templates that are currently made of polymer resin. Although few adverse effects have been recorded, the metal cylindrical sleeve in the guide template may separate from the resin guide template or the guide template may break. In this study, three cylindrical sleeves in the SLA group fell off. This can result in a deviation in the access cavity and lead to complications. One study has shown that using shorter sleeves can significantly improve accuracy (El Kholy et al., 2019). Similarly, reducing the height of the guide template for posterior teeth makes clinical procedures more convenient in limited operating spaces. The sleeve height of previous resin guide templates were mainly 6 mm (Krastl et al., 2016) or 8 mm (Lara-Mendes et al., 2018). In contrast, the SLM technology used in this study limits the height of the sleeve to within 5 mm, reducing the risk of sleeve detachment and improving accuracy. For better clinical application, the height of the SLA guide template was also reduced to 5 mm in this study. However, this may result in poor mechanical strength of the SLA guide template, leading to loosening and detachment of sleeves. The research by Gonçalves et al. (2021) on guided endodontics reduced the sleeve height to 4.1 mm, considering the thickness of the guide template itself, the height to the tooth surface was greater than 5 mm. The study by Buchgreitz, Buchgreitz & Bjørndal (2019b) used a plate of reduced height but only a single sleeve. This is not suitable for teeth with multiple calcified root canals. Moreover, thicker templates can make the patient feel uncomfortable (Buchgreitz, Buchgreitz & Bjørndal, 2019b). The resin guide template like SLA is usually weaker and more fragile, as well as relatively thick (Han et al., 2019). However, the SLM guide template has a reduced thickness of up to 1 mm (Lu et al., 2015), which minimizes patient discomfort and provides dentists with better visibility.

Owing to the limited space for the posterior guide template and drill (Connert et al., 2017; Fonseca Tavares et al., 2018), the application of guided endodontics has rarely been reported for posterior teeth. Most of the studies are case reports, lacking systematic research. In this study, a laser-melting template was designed to overcome the limitations of traditional templates and improve the effectiveness of posterior RCT by reducing the overall height of the template. Additionally, a systematic comparison was conducted between the SLM and traditional SLA guided templates to assess the deviations in posterior RCT and clarify the accuracy of the former. Other authors proposed a different posterior open and sleeveless template (Torres et al., 2021). The use of this template can improve visibility during the drilling process due to the lack of vertical space. However, the large size of the sleeveless template makes the patients feel uncomfortable, and prohibit the use of rubber dams during the operation. Compared to SLA, SLM structures have no internal porosity, exhibit a more uniform microstructure, and demonstrate better hardness and mechanical performance (Al Jabbari et al., 2019). Because of the superior mechanical strength of SLM, the connecting rods of the SLM template was appropriately lowered in this study. This caused the height of the SLM template to be lower than that of the SLA template, especially when applied to multiple sleeves.

In this study, the pathway deviation of the SLM group was significantly lower than that of the SLA group. High structural accuracy is also an advantage of SLM. The accuracy of 3D printing is affected by a combination of shrinkage, support structure, and build orientation, with shrinkage exerting the greatest influence (Calignano et al., 2016). Both SLM and SLA exhibited volumetric shrinkage during the manufacturing process. A study reported a volume shrinkage rate of 2.69% for epoxy resin and acrylic resin (Li et al., 2020), which is comparable to another study reporting a volume shrinkage rate of 2.91% (Wang, Habib & Zhu, 2018). In contrast, the dimensional shrinkage with SLM was less than 2.1%, which may improve accuracy and reduce wobbling of dental drills during insertion (Zahrul Adnan et al., 2014). The assembly clearance in the design of the guide template is for the smooth insertion of the metal sleeve. However, printing errors caused by factors such as resin shrinkage result in an unstable connection between the metal sleeve and the resin, resulting in slight wobbling of the metal sleeve during the drilling process. This may be a contributing factor to the inaccuracy of SLA and detachment of the metal sleeve. This is also an inevitable issue when manufacturing guide templates as separate components. Additionally, the tensile strength of 3D printed Co-Cr alloy is higher than photosensitive resin (911–951 MPa (Takaichi et al., 2013) vs. 49.6–52.2 MPa (Huang, Weng & Sun, 2011)), which may have a positive impact on stability. High tensile strength reduces deformation under insertion or pressure, providing a stable guiding direction.

Another factor affecting endodontic path preparation errors is the vibration of burs during the operation. In this study, the errors were significantly greater in the molars than in the premolars. Two access channels were designed for the molar guide template. Therefore, longer endodontic paths increased the deviation of molars, resulting in a significant reduction in deviation of overall posterior teeth and molars, but not of premolars. The longer paths in molars indicate that drill vibration affects the operational error, and the longer paths magnify this difference.

The resolution of CBCT affects the accuracy of root canal positioning (Fonseca Tavares et al., 2022). The resolution of traditional CBCT is 100–125 µm (Özer, 2011), while micro-CT has a resolution of 9–45 μm, indicating a need for further improvement in resolution (Ordinola-Zapata et al., 2017). Deviations in image processing, as well as the accuracy of intraoral scanning and software image fitting, can result in deviations in the final guide template. Some studies have used the common implant guide template design software “coDiagnostix” to design endodontic guide templates, which are complemented by CT and intraoral scanning images, yielding acceptable results (Loureiro et al., 2021; Zehnder et al., 2016). In this study, the average distance deviation range in the SLM group was 0.21 to 0.32 mm, and the average angle deviation range changed from 1.98° to 2.63°. According to the research by Zehnder’s research (Zehnder et al., 2016), the angle deviation range is 0~5.60°, with the deviation distance ranged from 0 to 1.59 mm, respectively. Another study on guided endodontics showed that the angle deviation ranged from 0° to 11.6° and the deviation from 0 to 1.60 mm (Su et al., 2021). Although no professional plate design software was used in this study, favorable experimental results were obtained compared to the previous studies.

However, the SLM template of guided endodontics has certain limitations. Dentists are unable to adjust the angle, depth, or type of dental drill during the treatment process, and therefore cannot adapt to unexpected issues that may arise during the surgery (Silva et al., 2022). It is a remarkable fact that dynamic navigation also has been used in endodontics for accessing obliterated root canals in some recent studies. The computer provides real-time feedback to the clinician during treatment (Jain, Carrico & Bermanis, 2020). Whereas the technique is highly dependent on operator experience and requires further learning for achieving mastery, which needs simultaneous hand–eye coordination. Static guidance is not operator-dependent and requires less treatment time than dynamic navigation (Kulinkovych-Levchuk et al., 2022). But both SLM and SLA templates require a lengthy design time (Connert, Weiger & Krastl, 2022). In contrast, the treatment outcomes of static-guided navigation are not affected by the operator’s experience (Connert et al., 2019). The technology is friendly to less-experienced dentists.

Conclusions

The SLM guide template has higher strength and lower height compared to the SLA guide template, which increases its stability and accuracy. These findings suggest their clinical application in the RCT of calcified posterior teeth, especially those with multiple calcified root canals. It also provides a better option for patients and dentists in managing PCO in posterior teeth. As this study is an in vitro study, there is still a gap between the findings and actual clinical application. Therefore, SLM guide template can be further optimized based on varying clinical cases.

Supplemental Information

Supplemental Information 1 Raw Data.

Additional Information and Declarations

Competing Interests

Author Contributions

Human Ethics

Data Availability

The authors declare that they have no competing interests.

Ting Zhang conceived and designed the experiments, performed the experiments, analyzed the data, authored or reviewed drafts of the article, and approved the final draft.

Du Chen conceived and designed the experiments, analyzed the data, authored or reviewed drafts of the article, and approved the final draft.

Fei Zhang conceived and designed the experiments, performed the experiments, authored or reviewed drafts of the article, and approved the final draft.

Sijing Xie conceived and designed the experiments, prepared figures and/or tables, authored or reviewed drafts of the article, and approved the final draft.

Guofeng Wu conceived and designed the experiments, prepared figures and/or tables, and approved the final draft.

Qingang Hu conceived and designed the experiments, authored or reviewed drafts of the article, and approved the final draft.

Fuhua Yan conceived and designed the experiments, authored or reviewed drafts of the article, and approved the final draft.

Xuna Tang conceived and designed the experiments, authored or reviewed drafts of the article, and approved the final draft.

The following information was supplied relating to ethical approvals (i.e., approving body and any reference numbers):

The Ethics Review Board of Nanjing Stomatological Hospital Medical School approval to carry out the study.

The following information was supplied regarding data availability:

The raw measurements are available in the Supplemental File.

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
