# Peer review of "Comparison of selective laser melting and stereolithography etching templates for guided endodontics"

_PeerJ, doi:10.7717/peerj.17646_

## Round 0.1 · original submission · Major Revisions

Dear Authors,

Firstly, I commend you on the dedication and diligence you've demonstrated in crafting this manuscript. It's evident that considerable time and effort have been invested in its development. I am pleased to inform you that your manuscript has undergone thorough evaluation by three external peer reviewers.
I suggest incorporating the objective and hypothesis at the conclusion of the introduction section to enhance clarity and provide readers with a clear understanding of the study's aims.
Furthermore, it would be beneficial to present the inclusion and exclusion criteria in the materials and methods section as bullet lists. This will streamline the presentation of this crucial information for readers' comprehension.
Regarding the image acquisition process, please provide additional details to offer readers a comprehensive understanding of the methodology employed.
Additionally, it has come to my attention that there are several typographical errors scattered throughout the text. To ensure the manuscript's professionalism and readability, I recommend enlisting the expertise of a proficient English copyeditor. Specifically, ensure consistency in the formatting of numbers, such as using numerals for quantities exceeding ten. Moreover, consider standardizing the terminology by selecting either "root canal calcification" or "obliteration" for consistency throughout the manuscript.
I trust that these revisions will enhance the overall quality and clarity of your manuscript. Please do not hesitate to reach out if you require further clarification or assistance.

·

Basic reporting

- Clear, courteous and unambiguous language;

- All points in the article were well expressed and capable of full understanding;

- The structure is correct and well designed in accordance with the magazine’s standards;

- I suggest the inclusion or replacement of more current references in the introduction, although this topic fulfills its objective, which is to expose a gap in knowledge to be resolved;

- In the introduction: “Our team has previously used laser-melting templates to treat calcified root canals in the anterior teeth and achieved good results(Zhang et al. 2020).” It is very important to value the research group, but I suggest you write the same sentence in third person, since we are on the introduction topic. Suggestion: “models to treat calcified root canals in anterior teeth obtained good results by the same research group as this study”.

Experimental design

- The “problem” in question that this study aimed to solve is well defined and explained;

- The methodology is well designed and reproducible;

- Material and methods: “80 premolars and molars were extracted for reasons unrelated to this study were collected” it would be interesting to mention the origin of these teeth.

Validity of the findings

- The conclusion responds to the problem that the authors sought to resolve at the beginning of the study, but I suggest adding the limitations of this study, since it was carried out in vitro;

- What are the next steps for models guided by selective laser fusion to reach clinical practice.

Additional comments

I suggest adding image credits, the name of who took the photographs and how the images were obtained in the caption.

Reviewer 2 ·

Basic reporting

The manuscript “Comparison of selective laser melting and stereolithography etching templates for guided endodontics” presents an interesting topic. The article includes sufficient introduction and background to demonstrate how the work fits into the broader field of knowledge.
Please pay attention:

1. RAW DATA: not appropriate according to the journal's standards

Experimental design

Some data from the methodology was not very clear.
1. What was the study's sample size? In line 81 it is described that "80 pre-molars and molars were extracted" but it is not clear how many are pre-molars and how many are molars. How did the authors arrive at the final n of 32 (16 pre-molars and 16 molars)?
2. There are only 9 molars with calcified pulp chambers, apart from the difficulty of access, calcified canals are not part of the study question? Shouldn't all the teeth selected have calcification?
3. Line 103 states that "The 3D models were then fitted with the intraoral scanned images", but at no point has it been previously described that _intraoral_ images were obtained, which images are these?
4. In "Traditional Plate", the information is redundant because the same sentence is repeated in line 118 and line 120. I suggest rewording the sentences.
5. In line 136, the first sentence of "Root canal pathway preparation" is unclear, "until no movement and displacement were detected between the molds and the teeth", what would this movement and displacement be?
6. It is also not described how the patient's bite was simulated, if at all. One question in the study is the difficulty of access in posterior teeth using guided endodontics, how was the height of the mouth opening simulated?
7. Figure 1 is not cited at any point in the text.

Validity of the findings

The results seem promising, despite the study's limitations.
On lines 247 and 248 it is described that the height of the guide makes patients feel uncomfortable, and a rubber dam cannot be used during the procedure. How could the technique be applied clinically? What suggestions for improving the technique do the authors suggest for future studies?

·

Basic reporting

The article is well grounded scientifically, bringing an innovative question. However, there are errors throughout the text, such as repeated sentences in the methodology, and the writing itself is not sufficiently robust and is difficult to interpret. Regarding the attached figures, Figure 1 is included, but there is no reference to it anywhere in the text.

Experimental design

The research question is well defined, I believe it to be relevant and innovative, since guided access in endodontics is a growing area and must be taken into account due to the difficulty in resolving some cases due to access to root canals.
The methodology contains repeated phrases throughout the text, in addition to being difficult to interpret, and some of the figures from the experiment can be cited for better understanding by the reader. Furthermore, there is no basis for choosing the number of teeth, with no sample calculation or previous studies to arrive at the number of teeth chosen for the study.

Validity of the findings

The results are well explained, however during the discussion the author mentions the difficulty of clinical applicability due to the height of the device and, therefore, difficulty in viewing and accessing the root canals. Furthermore, they report that due to the size of the guide, the patient may feel discomfort during the procedure and may not accept the procedure as well. Another important factor is the fact that absolute isolation cannot be performed during treatment, so the conclusion that suggests clinical applicability is hasty, given that there are still gaps to be studied.

Additional comments

There are some gaps in the study, such as a lack of clarity in the methodology, especially in the choice of teeth and the number of specimens. Furthermore, the device is proposed for teeth that present some type of calcification and not all specimens had this anatomy.
Methodology could be better written.

---

## Round 0.2 · accepted · Accept

Dear Authors,

I am pleased to inform you that your article has been accepted for publication. All the reviewers' comments and questions have been thoroughly addressed, and I have no further comments at this time. Thank you for your valuable contribution.

Best regards,

·

Basic reporting

None.

Experimental design

None.

Validity of the findings

None.

Additional comments

The authors carefully followed all the considerations made previously.